# Fast Joule Heating for the Scalable and Green Production of Graphene with a High Surface Area

**DOI:** 10.3390/ma17030576

**Published:** 2024-01-25

**Authors:** Zakhar Ivanovich Evseev, Aisen Ruslanovich Prokopiev, Petr Stanislavovich Dmitriev, Nikolay Nikolaevich Loskin, Dmitrii Nikolaevich Popov

**Affiliations:** Institute of Physics and Technologies, North-Eastern Federal University, 677000 Yakutsk, Russia; ar.prokopyev@s-vfu.ru (A.R.P.); dm.ni.popov@s-vfu.ru (D.N.P.)

**Keywords:** graphene, mildly oxidized graphene, electrochemically exfoliated graphene, fast joule heating, activated carbon, specific surface area

## Abstract

The rapid development of electric vehicles, unmanned aerial vehicles, and wearable electronic devices has led to great interest in research related to the synthesis of graphene with a high specific surface area for energy applications. However, the problem of graphene synthesis scalability, as well as the lengthy duration and high energy intensity of the activation processes of carbon materials, are significant disadvantages. In this study, a novel reactor was developed for the green, simple, and scalable electrochemical synthesis of graphene oxide with a low oxygen content of 14.1%. The resulting material was activated using the fast joule heating method. The processing of mildly oxidized graphene with a high-energy short electrical pulse (32 ms) made it possible to obtain a graphene-based porous carbon material with a specific surface area of up to 1984.5 m^2^/g. The increase in the specific surface area was attributed to the rupture of the original graphene flakes into smaller particles due to the explosive release of gaseous products. In addition, joule heating was able to instantly reduce the oxidized graphene and decrease its electrical resistance from >10 MΩ/sq to 20 Ω/sq due to sp^2^ carbon structure regeneration, as confirmed by Raman spectroscopy. The low energy intensity, simplicity, and use of environment-friendly chemicals rendered the proposed method scalable. The resulting graphene material with a high surface area and conductivity can be used in various energy applications, such as Li-ion batteries and supercapacitors.

## 1. Introduction

Porous carbon materials (PCMs), due to their structural features and chemical properties, have a wide range of applications [1] and have attracted significant attention from the scientific community [2]. The development of methods for fast, environmentally friendly, and inexpensive production of PCMs is especially relevant in connection with the rapid development of the energy industry for electric vehicles, unmanned aerial vehicles, and wearable electronics devices. Recently, great progress has been made in the production of synthetic carbon nanomaterials, such as carbon nanotubes and graphene materials [3]. Due to their high thermal and electrical conductivities, these materials are considered promising for use in power supplies with high power densities [4]. A large amount of research has been carried out on the use of graphene-based materials for the production of PCMs. [2]. In addition, carbon nanomaterials, such as graphene oxide (GO), have been shown to be promising additives for modifying the mechanical properties of various materials, making large-scale production relevant [5,6]. Research related to GO has received much attention, owing to the low cost of its production, as well as the possibility of flexible modification [7]. GO is an oxidized form of graphene that contains a large number of functional groups in various configurations [8]. GO is an electric insulator [6]. The electrical conductivity of GO can be significantly increased by reduction, which removes oxygen functional groups [9]. Various methods for GO reduction have been proposed, such as thermal annealing [10], laser annealing [11], and microwave irradiation [12]. In terms of scalability, chemical methods using reducing agents, such as hydrazine (N_2_H_4_) [13] and sodium borohydride (NaBH_4_) [14], have been well established. However, chemical reduction requires lengthy treatment times, leads to the contamination of the GO by the reducing agent, and is often performed using toxic and hazardous agents [15]. In addition, the high content of oxygen groups in the GO makes deep reduction problematic, which negatively affects the electrical conductivity. Methods for the synthesis of mildly oxidized graphene (MOG), such as electrochemical exfoliation [16], are of great interest. This method makes it possible to obtain GO with a low oxygen content of up to 20% [16], compared to 50% for GO obtained using classical methods [8]. In addition, electrochemical exfoliation can significantly reduce the synthesis time and is highly scalable. It can also be performed using non-toxic materials with high water solubility, which contributes to environmental friendliness and improves the purity of the product. A study [17] showed that electrochemical exfoliation of graphite can be carried out using various inorganic salts. The salts containing SO_4_^−^ anions were the most effective. Na_2_SO_4_, owing to its low cost and toxicity, can be used for large-scale production. However, this method has some major shortcomings. The main disadvantage is the exfoliation of large graphite particles during synthesis [18]. When detached from the graphite electrode, these particles do not participate in the electrochemical process and reduce the yield of the few-layer graphene flakes. Also, during swelling and oxidation of the original electrode, an uneven distribution of the electric field occurs in the volume of the electrode [19]. In addition, the molding of graphite electrodes is a separate technological step in which various impurities in the form of various binders and contaminants can be introduced. To solve these problems, Achee et al. [18] proposed electrochemical exfoliation in a contained volume. The authors used a dialysis bag containing an electrode and a graphite powder. The use of a contained volume made it possible to increase the yield to 65%, and increased the quality of the resulting MOG. In addition, a scalable flow reactor prototype was proposed. However, in the proposed design, it is impossible to stir the graphite powder to ensure the uniform processing of the entire volume.

The next technological step in the PCM production is the activation process [20]. Upon activation, a micro- and nanoporous structure is created. This allows for high specific surface area, helping to increase adsorption or specific energy density in energy applications [21]. The most common methods for activating carbon materials are chemical and physical activations [20]. Both methods involve lengthy thermal treatment of the original carbon precursor with a strong alkali solution, such as KOH [20]. The main disadvantages of these methods are their high energy intensity, process duration, and the use of dangerous chemicals [20,21]. Recently, research on methods for processing carbon materials using fast joule heating has gained much interest [22]. This method can be used to activate oxidized carbon materials [23]. The fast joule heating technique involves the rapid reduction of oxidized material during flash heating, associated with a large discharge current. The intense release of oxygen groups occurs in the form of gaseous products, which leads to the formation of pores when the graphene flakes rupture [23]. For the application of this method, MOG is more suitable, due to its electrical conductivity, in contrast to GO obtained by the Hummers method, which is a dielectric material. In [23], the authors coated carbon fabric with a layer of GO. Electrical pulses with a duration of 50 ms were then applied to the fabric at a voltage of 30 V. After treatment with 20 pulses, the specific surface area of the composite material increased to 166 m^2^/g. By mixing the initial GO with KOH, the specific surface area increased to 974 m^2^/g. In [24], carbon fibers were activated via fast joule heating. It was shown that when pulses of 4 to 8 V were applied, pores formed on the surface of the fibers. Increased specific surface area had a positive effect on the electrochemical performance of the treated material. However, the specific surface area was not analyzed in this study. To the best of our knowledge, activation by fast joule heating has only been investigated in these studies, making additional research in this area relevant.

In this study, a novel reactor was developed for the electrochemical exfoliation of graphite in a closed, expandable volume, which addressed the issue of large particle exfoliation. The proposed reactor design also provided the possibility of stirring the processed powder to increase the uniformity of exfoliation. To activate the MOG, a fast joule-heating setup was developed, and optimal parameters were selected to maximize the increase in specific surface area. This study presents a cost-effective and green method for the quick production of PCMs with a high specific surface area without the use of chemical processing. Additionally, fast joule heating resulted in the simultaneous reduction of MOG and the increase in electrical conductivity. The BET method was used to study the specific surface area, revealing a substantial increase in the specific area of MOG after processing in the fast joule heating installation.

## 2. Materials and Methods

The synthesis of MOG was carried out in a reactor with two volumes positioned vertically to one another and separated by a polyethylene terephthalate membrane with 1 µm pores (Figure 1). The upper volume served as a container for the electrolyte and was equipped with flat gold electrodes with a surface area of 5 cm^2^. A 0.1 M solution of Na_2_SO_4_ (Rushim, Moscow, Russia) was used as an electrolyte. The lower volume was equipped with a movable piston with a flat gold electrode with a surface area of 4 cm^2^ fixed on its surface.

The graphite powder was processed through a number of steps. First, 2 g of flake graphite (Sigma Aldrich, St. Louis, MO, USA) was placed in a lower volume, which periodically replenished the electrolyte and mixed graphite powder by moving the piston. A positive voltage of +10 V was applied to the graphite, causing it to oxidize and exfoliate, resulting in a volume increase in the original graphite powder. The exfoliation process took 6 h, after which the small particles of oxidized graphite were collected for further treatment. The obtained product was then rinsed using a vacuum filtration unit with a copious amount of deionized water. The resulting dry residue was mixed with 100 mL of deionized water to form an aqueous suspension of oxidized graphite. The suspension was treated with ultrasound at 60 W for 3 h using an Up 200St installation (Hielscher Ultrasonics, Teltow, Germany), resulting in the peeling of oxidized graphite particles and the formation of a suspension of the few-layer MOG. The MOG suspension was then centrifuged using Eppendorf MiniSpin plus (Eppendorf AG, Hamburg, Germany) at 14,500 rpm for 10 min to separate the unexfoliated particles by decanting it over the sediment. To obtain MOG powder, freeze-drying was performed using the vacuum freeze-dryer Biobase Scientz-10N (Ningbo Scientz Biotechnology Co., Ltd., Zhejiang, China).

To activate the MOG, the methodology outlined in a previous study [25] and the studies discussed in [22] were employed. A 32 mF capacitor bank (Epcos AG, Munich, Germany) was charged to the required value and then discharged into the reaction chamber at voltages ranging from 100 to 220 V. The reaction chamber consisted of two copper electrodes that pressed the MOG powder in a quartz tube (Dominik Co., Moscow, Russia) with a diameter of 14 mm. The discharge occurred in a nitrogen atmosphere at a pressure of 0.3 mbar and took approximately 32 ms. The activated material was labeled activated MOG (aMOG), and the samples corresponding to the processing voltages were aMOG–100, aMOG–140, aMOG–180, aMOG–200, and aMOG–220.

The Raman spectra of the resulting material were examined using the NTegra Spectra installation (NT–MDT, Zelenograd, Russia). The surface morphology of the resulting material was analyzed using scanning electron microscopy (SEM) with a JEOL-7800F microscope (Jeol, Tokyo, Japan). Atomic force microscopy (AFM) was employed to study the individual MOG and aMOG flakes using Solver Next (NT–MDT, Zelenograd, Russia). The elemental composition was determined using X-ray energy-dispersive spectroscopy (EDS) with a NanoAnalysis microanalysis system (Oxford Instruments, Oxford, UK) attachment of the SEM. The functional composition was studied through Fourier-transform infrared spectroscopy (FTIR) with a Spotlight 200i spectrometer (PerkinElmer, Waltham, MA, USA). Measurements of current–voltage characteristics (C–V) were performed using a two-probe method on ASEC-03 (Prokhorov General Physics Institute of the Russian Academy of Sciences, Moscow, Russia) and AMM-3046 (Aktakom, Moscow, Russia) in the voltage range of −1 to +1 V. The specific surface area was determined using the Brunauer–Emmett–Teller method on NOVAtouch LX (Quantachrome Instruments, Inc., Boynton Beach, FL, USA) with a static volumetric method to measure the amount of adsorbed nitrogen. The preparation of the samples was carried out in accordance with the ISO 9277:2022 standard [26]. The samples were degassed at a residual pressure of 10 μT and a degassing temperature of 350 °C for 12 h. The free space (“dead volume”) of the cells was initially determined using helium. The purities of N_2_ and He were >99.999 %.

## 3. Results

Figure 2 shows AFM images of the initial MOG (a) and aMOG–200 (b) individual flakes. Thickness profiles are displayed in each image. The average lateral dimensions of individual MOG flakes were in the range between 0.05 and 0.8 µm, with thicknesses varying between 6 and 20 nm. In comparison, the lateral dimensions of the aMOG–200 flakes were within 100 nm, with thicknesses of 0.8 to 1.2 nm. On average, individual flakes of MOG exhibited a greater number of graphene layers than aMOG, as well as larger lateral dimensions. This indicates that the initial MOG flakes underwent rupture during the activation process with fast joule heating.

Images of the MOG and aMOG–200 film surfaces were obtained using the SEM (Figure 3). It is shown that the MOG film consisted of agglomerated particles formed during the drying of the MOG suspension. The surface of aMOG–200 had a micro- and nanoporous structure formed as a result of activation.

The elemental analyses of the initial MOG and aMOG–200 are presented in Table 1. It should be noted that the EDS did not allow the identification of hydrogen atoms. The data indicate that the developed reactor successfully synthesized MOG with a low oxygen content of 14.1%. MOG was reduced by rapid joule heating to an oxygen content of 4.2%.

Figure 4a shows the FTIR spectra of the MOG and aMOG–220. The peak localized in the vicinity of 1250 cm^−1^ corresponds to the O–H hydroxyl group [27,28]. The presence of the hydroxyl group was confirmed by the presence of vibrations in the region of 3200–3400 cm^−1^, also associated with O–H groups and H_2_O [28]. The peak associated with stretching vibrations of carboxyl groups (–COOH) was observed in the region of 1725 cm^−1^. The disturbances localized in the vicinity of 1091 cm^−1^ were associated with the presence of epoxy groups (C–O–C). Stretching vibrations of C–O bonds of alkoxy groups in the region of 1044 cm^−1^ were present [27]. It is worth noting that for aMOG, the C–O bonds (1131 cm^−1^) associated with carbonyl groups [27,28] were absent. The presence of C=C bonds was also observed (1680–1710 cm^−1^), which are related to the vibrations of sp^2^ crystallites of graphite [27,28].

Figure 4b shows the Raman spectra of MOG and aMOG–220 samples obtained after joule heating. All powders were characterized by the presence of the main peaks, D (1350 cm^−1^) and G (1580 cm^−1^), corresponding to graphite- and graphene-containing structures [29]. Raman spectra of MOG samples demonstrated the presence of a wide-band 2D peak in the region of 2700–3100 cm^−1^. After the fast joule heating, the Raman spectra of aMOG–220 changed significantly. The D peak intensity decreased. A clearly defined peak at 2701 cm^−1^, corresponding to the second-order 2D peak, was observed [30]. The intensity of the 2D peak shows the graphitization of the initial MOG, which, coupled with the low intensity of the D peak and the sharp G peak, indicates the formation of ordered graphene flakes [31].

The Raman spectra of aMOG obtained under various discharge voltages are shown in Figure 5a. Peaks corresponding to graphite- and graphene-containing structures (D, G, 2D, and D + G) were identified [32]. The first-order Raman peaks, called the D and G peaks in the Raman spectra, are associated with the disordering of the lattice and vibrations of sp^2^-hybridized carbon bonds [33], respectively. The 2D (2700 cm^−1^) peak of the Raman spectra corresponds to the overtone of the D peak [34], which represents the presence of graphene layers [29]. The band located at ~2900 cm^−1^ (D + G) is a combination of the D and G peaks and is also associated with defects [34,35]. As can be seen from Figure 5a, the peak at 3150 cm^−1^ disappeared with increasing voltage. This peak corresponds to the hydroxyl and carbon–hydrogen groups [36]. An increase in the energy of discharge contributed to the removal of peaks localized in the frequency range between 2900–3100 cm^−1^. Figure 5b shows the decomposition of the Raman spectra of aMOG–220 into Lorentzian peaks. The ratio of the integrated peak intensities (I_D_/I_G_), which is responsible for assessing the disorder of the carbon structure, was 0.45, which corresponds to the defective graphene [37]. From the empirical formula [38] defined by Cancado et al., the lateral dimensions of sp^2^ crystallites of nanographite were calculated (Table 2). The ratio of I_2D_/I_G_ shows that the aMOG–220 was similar to few-layer graphene. The estimate at the first approximation was up to 7 layers [39], which correlates with the data obtained by the AFM method.

Table 2 shows the values of the surface resistance (R/sq.) of the aMOG, obtained from measurements of C–V characteristics using the two-probe method.

Figure 6 shows the average values of the surface resistance, depending on the treatment voltage. The surface resistance values were plotted as columns, and the estimated lateral sizes of sp^2^ crystallites were plotted as a line. It can be seen that with increasing voltage, R decreased from the initial values (from the insulator) to tens of Ω/sq., indicating the reduction of MOG and regeneration of the graphene structure, which is confirmed by the growth of the sp^2^ crystallites L_a_. Due to the high degree of disorder in the initial MOG and aMOG–100, L_a_ was not assessed. With an increase in the lateral dimensions of the sp^2^ crystallites, the electrical conductivity of aMOG increased by up to four orders of magnitude.

The specific surface areas measured using the BET method are listed in Table 3. From the calculated values, it can be concluded that with increasing discharge voltage, the specific surface area also increased to a maximum at 200 V—1984.5 m^2^/g. However, at the discharge voltage of 220 V, a sharp decrease in the surface area was observed.

BET studies showed that the adsorption isotherms for all samples, according to the IUPAC classification, belonged to type IV. The hysteresis loops in all isotherms were of the H1 type, which is characteristic of agglomerates of spherical particles that are uniformly packed and similar in size. The nature of the adsorption branches at low relative pressures indicated the presence of a certain number of micropores. All the samples exhibited a multimodal distribution of pore size in the range of 2–10 nm. The pore size distribution was correlated with the thicknesses of the aMOG samples measured by AFM. The distribution of the pores by surface area of the aMOG–200 with the highest measured specific surface area is presented in Figure 7b. It is shown that the pore distribution had maxima at 2.8 and 3.7 nm.

## 4. Discussion

The results obtained from the data indicate that the reactor developed in this study can produce multilayer flakes of MOG with an oxygen content up to 14.1%. The composition of the functional groups of the initial MOG correlates with the data obtained in other studies [16]. At the same time, the reactor design has the possibility of significant scaling. The use of the fast joule heating method made it possible to increase the conductivity of the aMOG by four orders of magnitude, which can be associated with the thermal reduction of the initial MOG to an oxygen content of 4.2%. Raman spectra analysis showed an increase in sp^2^ crystallite size L_a_ with the increase in fast joule heating voltage, which is an indication of the reduction of MOG and the regeneration of the sp^2^ carbon lattice. Simultaneously, the explosive release of oxygen groups in the form of gaseous products during fast joule heating (32 ms) led to the rupture of the initial MOG individual flakes [23]. AFM studies showed decreases in lateral dimensions from 0.8 μm to 100 nm and thicknesses from 6–20 nm to 0.8–1.2 nm for aMOG, compared to the initial MOG. This explains the significant increase in the specific surface area of aMOG–200 to 1984.5 m^2^/g at a processing voltage of 200 V. The hysteresis loops of the adsorption isotherms of the material were of type H1, which is characteristic of agglomerates of spherical particles that are uniformly packed and similar in size. At the same time, the Raman spectra showed a decrease in the D peak associated with structural defects. This could indicate that percolation did not occur in the lateral plane of the flakes during activation by fast joule heating. The mechanism of flake rupture appears to be the delamination of the multilayer graphene into thinner flakes and lateral cracking into smaller particles. The data obtained from BET analysis indicated a significant increase in the specific surface area as the processing voltage was increased to a maximum area at 200 V. At a processing voltage of 220 V, the increase in the specific area was significantly lower. It can be speculated that this effect may be associated with a regeneration of the sp^2^ structure, which leads to the restacking of individual flakes via the Van der Waals forces. This is indirectly confirmed by the high electrical conductivity of aMOG–220 (~20 Ω/sq). In addition, the pore size distribution in aMOG–220 shifted towards a multimodal distribution in the range >5 nm, compared to the <5 nm distribution in aMOG–200, which can be attributed to the restacking of the flakes. Additional research is required to determine the reasons for this effect.

In conclusion, an original reactor was developed for the electrochemical synthesis of MOG. The synthesis of MOG was carried out without the use of toxic and dangerous agents and also had low energy intensity, which is the basis for possible scalability. To activate the synthesized MOG and obtain graphene-based PCMs, fast joule heating was used. This method made it possible to obtain graphene-based PCMs with a high specific surface area of up to 1984.5 m^2^/g. Due to its speed, simplicity, and low energy consumption, the developed technique can be used for the green production of PCMs for various energy sources, such as Li-ion batteries and supercapacitors.

## Figures and Tables

**Figure 1 materials-17-00576-f001:**
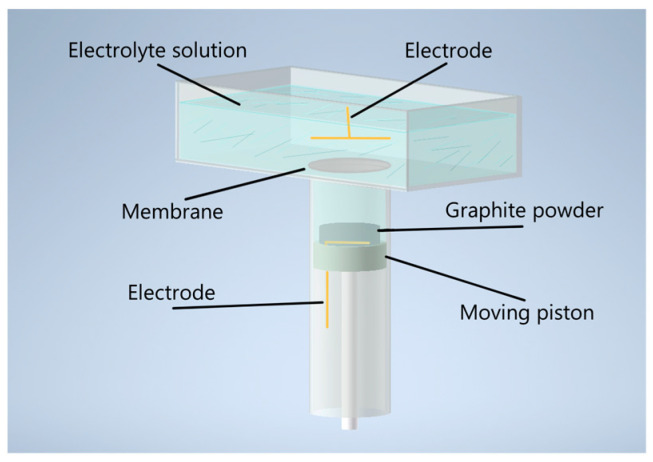
Schematic representation of the electrochemical reactor for MOG synthesis.

**Figure 2 materials-17-00576-f002:**
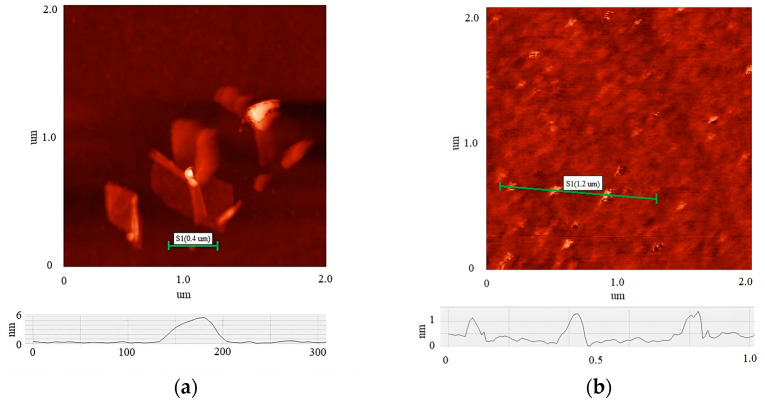
AFM images of individual flakes: (**a**) MOG; (**b**) aMOG–200. The height profiles of the flakes are shown at the bottom.

**Figure 3 materials-17-00576-f003:**
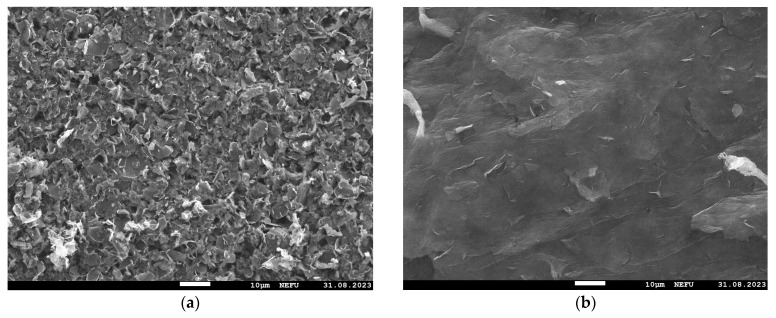
SEM images at 1000× magnification: (**a**) MOG; (**b**) aMOG–200.

**Figure 4 materials-17-00576-f004:**
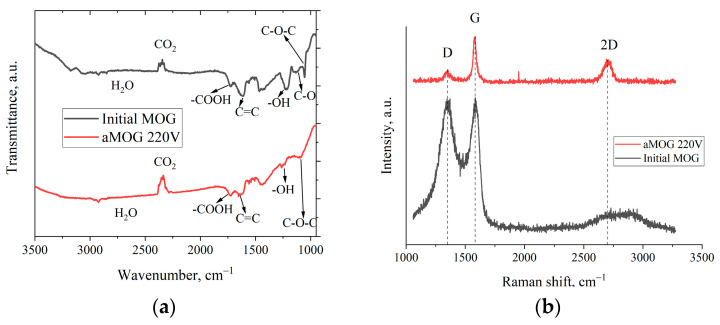
(**a**) FTIR spectra of MOG and aMOG–220; (**b**) Raman spectra of the MOG and aMOG–220.

**Figure 5 materials-17-00576-f005:**
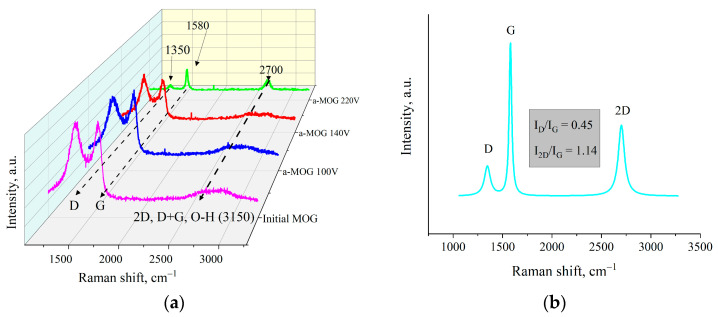
(**a**) Raman spectra of samples obtained after fast joule heating of the MOG at different voltages; (**b**) Lorentzians of the aMOG–220 Raman spectra.

**Figure 6 materials-17-00576-f006:**
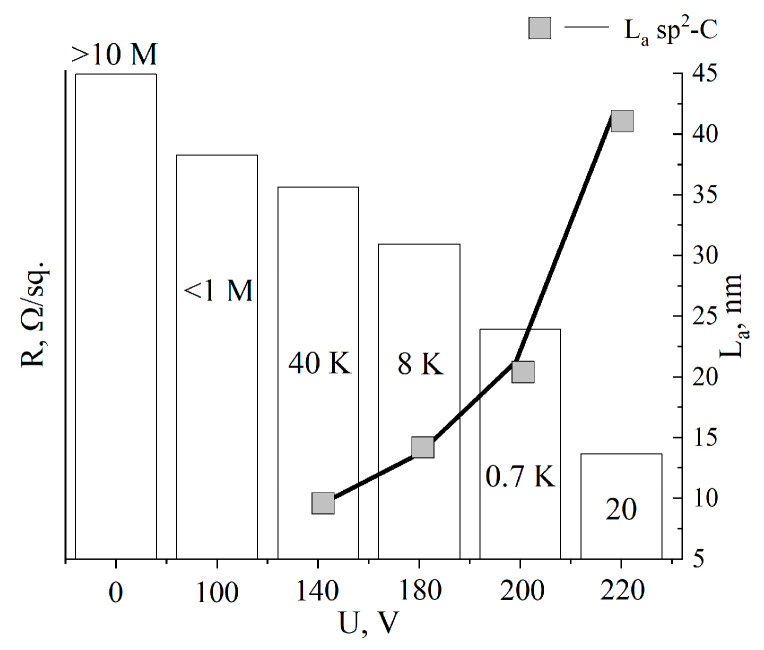
Histogram of R/sq. and L_a_ distributions depending on the voltage of joule heating. Rectangles—L_a_ values; text above and inside of the histogram—resistance values.

**Figure 7 materials-17-00576-f007:**
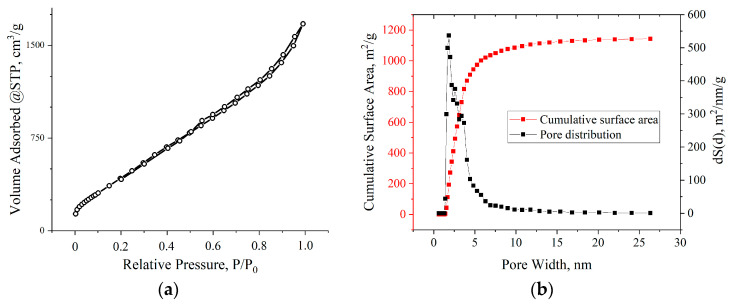
(**a**) Typical isotherm of nitrogen adsorption–desorption on aMOG samples; (**b**) distribution of the pore surface area of aMOG–200.

**Table 1 materials-17-00576-t001:** Carbon and oxygen contents in MOG and aMOG–200 determined by EDS.

Content	MOG	aMOG–200
C, at. %	85.9	95.8
O, at. %	14.1	4.2

**Table 2 materials-17-00576-t002:** Values of the surface resistance (R/sq.), ratio of Raman I_D_/I_G_ peaks, and lateral sizes of sp^2^ crystallites (L_a_), depending on the activation voltage.

Voltage, V	Resistance, R/sq	I_D_/I_G_	L_a_, nm
Initial MOG	>10 MΩ	>3	–
100	<1 MΩ	>3	–
140	40 kΩ	<2	9.6
180	8 kΩ	1.4	13.7
200	0.7 kΩ	0.9	21.3
220	20 Ω	0.45	42.6

**Table 3 materials-17-00576-t003:** Dependence of the specific surface area on the discharge voltage.

Voltage, V	S_BET_, m^2^/g
Initial MOG	181.5
100	278.2
140	313.3
180	414.6
200	1984.5
220	260.5

## Data Availability

The data presented in this study are available upon request from the corresponding author. The data are not publicly available due to the data protection policy of the university.

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
