# Peer review of "Fast Joule Heating for the Scalable and Green Production of Graphene with a High Surface Area"

_materials, 2024, doi:10.3390/ma17030576_

Round 1

Reviewer 1 Report

Comments and Suggestions for Authors

The work presents a new synthesis method based on the electrochemical oxidation process of graphite without using chemical agents, which helps obtain a product with fewer impurities. It is necessary to answer the following questions to complete the information in this work.

What type are the gold electrodes in the reactor?

What surface area do gold electrodes have?

What is the speed at which the piston should move?

What is the degree of conversion of graphite to graphene using the synthesis method described in this research?

Author Response

Thank you for your review. It is great opportunity for us to receive feedback from the world scientific community.

1) What type are the gold electrodes in the reactor? What surface area do gold electrodes have?

We have revised section with reactor description in lines 112-118:

The synthesis of MOG was carried out in a reactor with two volumes positioned ver-tically to one another and separated by a polyethylene terephthalate membrane with 1 µm pores (Fig. 1). The upper volume served as a container for the electrolyte, and was equipped with flat gold electrodes with a surface area of 5 cm2. A 0.1 M solution of Na2SO4 (Rushim, Moscow, Russia) was used as an electrolyte. The lower volume was equipped with a movable piston with a flat gold electrode with a surface area of 4 cm2 fixed on its surface.

2) What is the speed at which the piston should move?

We did not conduct a study on the influence of piston speed. The piston movement was intended to stir the graphite and was performed once every 10 min during the process. The piston was driven manually, and its speed was limited by the permeability of the membrane and strength of the laboratory assistant. In future studies, we plan to introduce a mechanical drive for the piston. Thank you for the suggestion.

3) What is the degree of conversion of graphite to graphene using the synthesis method described in this research?

The disadvantage of the reactor in this variant is the manual movement of the piston. As the process progressed, exfoliated graphite formed a film on the surface of the membrane, which significantly increased the force required to move the piston. Thus, the yield was limited by the muscular strength of laboratory staff. Therefore, we did not include data on reactor yield. As mentioned previously, in the future, we plan to develop a reactor with a mechanical drive and more stable structure, which could theoretically increase the conversion rate to 100%.

Reviewer 2 Report

Comments and Suggestions for Authors

The authors described the work entitled “Fast joule heating for scalable and green production of graphene with a high surface area”. This work is well written, some images can be optimized in terms of quality (such as the AFM profiles..) but in general it could be accepted after some minor revisions. My comments are reported below:

(1)   From this method, is it possible to monitor the Graphene thickness?

(2)   Regarding the crystallite lateral size (La), it is necessary to explain in the text the correlation between the Raman peaks and the grain dimensions. I suggest the following papers:

a.       https://doi.org/10.1016/j.carbon.2009.07.033

b.      https://doi.org/10.1103/PhysRevB.91.195411

c.       https://doi.org/10.1088/1361-6528/abb72b

(3)   What is the physical reason for the decreasing in specific surface area at 220 V?

(4)   Figure 6 and 7 need to be described better.

Author Response

Thank you for your review. Working on the problems that you identified allowed us to improve our paper. It is great opportunity for us to receive feedback from the world scientific community.

 (1)   From this method, is it possible to monitor the Graphene thickness?

Thank you for your comments. It would be interesting to conduct a series of experiments to establish the relationship between the electrolyte concentration, processing voltage, and process duration with the thickness of the resulting midly oxidized graphene. In future studies, we will examine this relationship. At this stage, the thickness of the resulting graphene depended mainly on the centrifugation step. The more intense the centrifugation, the lower the average thickness but with a significant decrease in the yield. In this study, the optimal parameters were selected to obtain flakes with a thickness of 6-20 nm at 14500 rpm for 10 min.

(2)   Regarding the crystallite lateral size (La), it is necessary to explain in the text the correlation between the Raman peaks and the grain dimensions. I suggest the following papers:

  1. https://doi.org/10.1016/j.carbon.2009.07.033
  2. https://doi.org/10.1103/PhysRevB.91.195411
  3. https://doi.org/10.1088/1361-6528/abb72b

Thank you for your comments. We revised this part in lines 205-218:

The Raman spectra of aMOG obtained under various discharge voltages are shown in Figure 5a. Peaks corresponding to graphite and graphene–containing structures (D, G, 2D, and D+G) were identified [31]. The first-order Raman peaks, called the D and G peaks in the Raman spectra, are associated with the disordering of the lattice and vibrations of sp2-hybridized carbon bonds [32], respectively. The 2D (2700 cm-1) peak of the Raman spectra corresponds to the overtone of the D peak [33], which represents the presence of graphene layers [28]. The band located at ~2900 cm-1 (D + G) is a combination of the D and G peaks and is also associated with defects [33, 34]. As can be seen from Figure 5a, the peak at 3150 cm–1 disappears with increasing voltage. This peak corresponds to the hy-droxyl and carbon–hydrogen groups [35]. An increase in the energy of discharge contrib-utes to removal of peaks localized in the frequency range 2900, 3100 cm–1. Figure 5b shows the decomposition of the Raman spectra of aMOG–220 into Lorentzian peaks. The ratio of the integrated peak intensities (ID/IG), which is responsible for assessing the disorder of the carbon structure, is 0.45, which corresponds to the defective graphene [36].

(3)   What is the physical reason for the decreasing in specific surface area at 220 V?

We clarified the explanation in the discussion part. Lines 280-287:

At a processing voltage of 220 V, the increase in the specific area was significantly lower. It can be speculated that this effect may be associated with a regeneration of the sp2 struc-ture, which leads to the restacking of individual flakes via the Van–der–Waals forces. Which is indirectly confirmed by the high electrical conductivity of aMOG–220 (~20 Ω/sq). In addition, the pore size distribution in aMOG–220 shifted towards a multimodal distri-bution in the range of >5 nm, compared to the <5 nm distribution on aMOG–200, which can be attributed to the restacking of the flakes. Additional research is required to deter-mine the reasons for this effect.

(4)   Figure 6 and 7 need to be described better.

Thank you for the remark we revised descriptions of both images

Reviewer 3 Report

Comments and Suggestions for Authors

    I read the contribution “Fast joule heating for scalable and green production of graphene with a high surface area” with interest. This study introduced a reactor designed for environmentally friendly, straightforward, and scalable electrochemical synthesis of graphene oxide, yielding a material with a low oxygen content. Subsequently, the resulting material underwent activation through a rapid joule heating process. This work holds substantial importance for publication as a scientific paper in a reputable journal. However, at its current stage, it requires several essential modifications and clarifications before a final decision regarding rejection or acceptance can be made. The following outlines the most prominent amendments needed.

1.      The Abstract section needs improvement as it doesn't offer a complete overview of the study

2.      Graphene oxide has been shown to enhance the mechanical properties of various materials. This improvement merits discussion in the introduction, referencing relevant articles e.g. a) Strategic formulation of graphene oxide sheets for flexible monoliths and robust polymeric coatings embedded with durable bioinspired wettability. ACS applied materials & interfaces 9 (48), 42354-42365. b) Robust and self-healable bulk-superhydrophobic polymeric coating. Chemistry of Materials 29 (20), 8720-8728.

3.      Do alternative inorganic salt-based electrolytes, aside from Na2SO4, impact the process of anodic exfoliation in graphite?

4.      Authors are advised to incorporate a comparative table illustrating the recent advancements in electrochemical synthesis graphene oxide and , highlighting the position of their own research within this field.

5.      The authors are encouraged to improve Figure 1 as the electrolyte solution is not clearly visible in its current presentation.

6.      What are the dimensions of both MOG and MOG-200?

7.      In order to gain a comprehensive understanding of the chemical composition, could the author provide XPS data for both MOG and MOG-200?

8.      Why does the intensity peak of C=C in the FTIR spectra of aMOG-220 decrease compared to MOG?

Comments on the Quality of English Language

Minor editing of English language required

Author Response

Thank you for a review of our study, as well as for pointing out the shortcomings in our study. We hope that the work on the points that you noted allowed us to improve our paper. It is great opportunity for us to receive feedback from the world scientific community.

Round 2

Reviewer 3 Report

Comments and Suggestions for Authors

Thank you for your response; all the inquiries have been thoroughly answered.